# The Roles of Osteopontin in the Pathogenesis of West Nile Encephalitis

**DOI:** 10.3390/vaccines8040748

**Published:** 2020-12-09

**Authors:** Farzana Nazneen, Fengwei Bai

**Affiliations:** Department of Cell and Molecular Biology, Center for Molecular and Cellular Biosciences, The University of Southern Mississippi, Hattiesburg, MS 39406, USA; Farzana.Nazneen@usm.edu

**Keywords:** osteopontin, West Nile virus, encephalitis, pathogenesis

## Abstract

Osteopontin (OPN), a multifunctional protein encoded by the *secreted phosphoprotein-1 (Spp-1)* gene in humans, plays important roles in a variety of physiological conditions, such as biomineralization, bone remodeling and immune functions. OPN also has significant roles in the pathogenesis of autoimmune, allergy and inflammatory diseases, as well as bacterial, fungal and viral infections. West Nile virus (WNV), a mosquito-transmitted flavivirus, is the leading agent for viral encephalitis in North America. Recent progress has been made in understanding both the biological functions of OPN and the pathogenesis of WNV. In this review article, we have summarized the current understanding of the biology of OPN and its vital roles in the pathogenesis of WNV encephalitis.

## 1. Osteopontin

### 1.1. The Biology of Osteopontin

Osteopontin (OPN) is a negatively charged, acidic and reversibly phosphorylated adhesive glycoprotein [1]. It is also known as bone sialoprotein-1 (BSP-1), early T-lymphocyte activation-1 (Eta-1), or *secreted phosphoprotein-1 (Spp-1)* [2]. OPN is a member of the non collagenous protein group called small integrin binding ligand and N-linked glycoprotein (SIBLING), which also includes dentin matrix protein-1 (DMP-1), dentin sialophosphoprotein (DSPP), bone sialoprotein-2 (BSP-2) and matrix extracellular phosphoglycoprotein-1 (MEPE-1) [3,4]. The human OPN gene, *Spp-1* is located on chromosome 4 (4q22.1) at the end of the SIBLING family [5] (Figure 1). The expression of *Spp-1* is regulated by multiple factors including cytokines (e.g., interleukin-1β (IL-1β), IL-6, tumor necrosis factor-α (TNF-α) and interferon-γ (INF-γ)), platelet-derived growth factor, oxidized low-density lipoprotein and hormones (e.g., vitamin D, estrogen, angiotensin II and glucocorticoids) [5,6]. OPN is expressed at high levels in human bone, joints, lung, liver, brain, adipose tissues and body fluids including blood, urine and milk [7,8,9]. Although OPN exists both as an immobilized extracellular matrix (ECM) molecule in mineralized tissues and as a cytokine in body fluids, it is not a typical nonmineralized ECM [10,11]. OPN is produced by a variety of cell types, such as bone cells (e.g., osteoblasts, osteoclasts and osteocytes), immune cells (e.g., T cell, B cell, natural killer cell, dendritic cell (DC) and macrophage), neural, epithelial, fibroblasts, smooth muscle and endothelial cells [5]. OPN acts as a ligand for several integrins, such as α_5_β_1_, α_8_β_1_, α_v_β_1_, α_v_β_3_, α_IIb_β, α_v_β_5_, α_v_β_6_, α_9_β_4_, α_9_β_1_ and α_4_β_1_ [12,13,14]. It is also a ligand for the cluster of differentiation 44 (CD44) molecules specifically for v6 and v7 classes providing antiapoptotic signals [15,16]. Through the receptor–ligand interactions, OPN exhibits diverse functions in promoting cancer cell survival, tumor formation, metastasis, granuloma formation, dystrophic calcification and coronary restenosis [10,16,17].

### 1.2. The Different Isoforms of OPN

*Spp-1* consists of seven exons that can produce three different splice variants, i.e., OPN-a, OPN-b and OPN-c through alternative RNA splicing (Figure 2). While the mRNA of OPN-full-length (OPN-FL, OPN-a) contains all of the seven exons, the other splice variants, such as OPN-b and OPN-c, lack exon-5 or exon-4, respectively [5]. All OPN mRNAs contain exon-2, which acts as the signal sequence. The OPN-FL/OPN-a mRNA can be translated into two functionally distinct isoforms, secreted OPN-a (sOPN) and intracellular OPN-a (iOPN), respectively. Both OPN-FL isoforms are generated from two alternative translation initiation sites, the start codon, adenine uracil guanine (AUG) site with the signal sequence or an alternate non-AUG site without the signal sequence [5]. The AUG site initiates translation to produce OPN-a (sOPN) isoform consisting of an N-terminal signal sequence allowing it to be secreted outside the cell and functioning as a pleiotropic cytokine [5]. The alternative non-AUG site initiates translation to generate OPN-a (iOPN), which remains in the cytoplasm, premembrane region and nucleus of DCs [18,19,20,21,22,23]. OPN-b mRNA and OPN-c mRNA produce the other two isoforms of sOPNs (OPN-b and OPN-c, respectively) and are translated from their mRNAs and secreted out from the cytoplasm [5].

OPN-a contains 314 amino acid (aa) residues, including a hydrophobic leader signal sequence of 16 aa [2]. OPN-a has one arginine–glycine–aspartic acid domain (RGD) with a cryptic sequence revealed upon cleavage by thrombin [24]. OPN also has several binding sites for integrins, CD44, calcium and heparin [5]. The RGD binding site is flanked by a 50-aa sequence, which is critical for binding to cell surface integrins to regulate cell attachment, spreading, migration and intracellular signaling [25]. The OPN-a (iOPN) interacts with interferon regulatory factor-7 (IRF-7) and induces IFNα expression in plasmacytoid dendritic cells (pDC) and promotes differentiation of IL-17 producing T helper cells (Th-17 cells) in conventional dendritic cells (cDC) [20,26,27]. OPN-a (iOPN) has also been shown to play a critical role in cell migration, fusion and motility. OPN-a (iOPN) can be colocalized with CD44 complex in cell processes, such as filopodia and pseudopodia, mediating cell migration through cytoskeletal rearrangement in fibroblastic cells and osteoclasts [28,29]. OPN-a (iOPN) is also involved in mitosis and progression of the cell cycle by influencing polo-like kinase-1 [23]. In Alzheimer’s disease, OPN-a (iOPN) is found to be involved in abnormal β-amyloid protein aggregates [23,30]. 

### 1.3. The Function of Special Forms of OPN

Functional diversity of OPN is usually regulated by several post translational modifications (PTMs), such as polymerization, matrix metalloproteinase (MMP) modification, thrombotic cleavage, phosphorylation, o-glycosylation, proteolytic processing and tyrosine sulfation and sialylation [31,32,33]. Here, we briefly discuss some common PTMs, such as polymerization and thrombotic cleavage.

#### 1.3.1. Polymerized OPN

OPN can be polymerized to form poly OPN by enzyme transglutaminase-2 (TG-2) [34]. TG-2 is a Ca^2+^-dependent protein cross-linking enzyme that catalyzes the formation of a covalent, γ-glutamyl-ε-lysyl (isopeptide) bond between specific Lys and Gln residues of its substrate proteins [35,36]. It is not yet clear if all of the OPN isoforms can undergo polymerization. However, the presence of poly OPN has been confirmed in bone [37] and aortic tissue of matrix Gla protein-null mice [38]. OPN acts as a chemotactic factor for neutrophils, lymphocytes and macrophages [39]. The polymerized OPN acquires some functional modifications including the increased chemotactic ability for neutrophils by binding with integrin α_9_β_1_ both in vivo and in vitro [40]. OPN polymerization in vivo is dynamically regulated and could contribute to the regulation of tissue inflammation. The increased number of TG-2 at the site of inflammation could lead to a rapid polymerization of OPN, resulting in an increased level of neutrophils in the inflammatory site [40]. 

#### 1.3.2. Thrombin-Cleaved OPN

OPN can be functionally and post translationally modified by thrombin cleavage. Two conserved thrombin cleavage sites have been identified in human OPN (Figure 3). The major thrombin cleavage site, termed Site-1, produces an N-terminal fragment containing the RGD domain termed OPN-N1 and a C-terminal fragment termed OPN-C1. The Site-2 cleavage exposes a cryptic site named the SVVYGLR (arginine–valine–valine–tyrosine–glycine–leucine–arginine) domain located within the RGD domain. Thrombin cleaves OPN at Site-2 and produces another distinct N-terminal fragment termed OPN-N2 containing the SVVYGLR domain and C-terminal fragment termed OPN-C2. 

OPN-C1 and OPN-C2 exert an effect on cell–cell adhesion by interacting with CD44 isoforms containing v6 and v7 domains and inhibit apoptosis [12]. Compared to C-terminal fragments, N-terminal fragments show more biological activity due to the presence of the RGD and cryptic SVVYGLR domains [12,41]. The RGD domain of OPN-N1 can bind to integrins α_v_β_1_, α_v_β_3_, α_v_β_5,_ α_v_β_6,_ α_5_β_1_ and α_8_β_1_ [12,13]. In contrast, the cryptic SVVYGLR domain of OPN-N2 acts as the ligand for integrins α_4_β_1_, α_4_β_7_, α_9_β_1_ and α_9_β_4_ to mediate cellular adhesion [13,25,42,43,44] (Figure 3). As α_4_β_1_ integrin is expressed on lymphocytes and smooth muscle cells, integrin α_4_β_1_ and SVVYGLR domain of OPN-N2 interaction may be involved in lymphocyte transmigration to the brain [12,13,44]. The SVVYGLR domain of OPN-N2 can also mediate neutrophil migration through binding with α_4_β_1_ integrin [45].

## 2. Roles of OPN in the Pathogenesis of WNV 

### 2.1. West Nile Virus

WNV is a single-stranded, positive-sensed, mosquito-transmitted RNA virus belonging to the family of *Flaviviridae*. WNV infection in humans is asymptomatic in the majority of individuals; however, in some cases, it can cause symptoms ranging from fever to neurological complications, such as meningitis, encephalitis, flaccid paralysis and even death. WNV was first discovered in Africa and first transmitted in the western world in 1999. WNV is now causing the most cases of mosquito-borne encephalitis in North America [46].

### 2.2. OPN and WNV

The OPN-FL protein plays various roles in the pathogenesis of WNV encephalitis, such as promoting inflammation [47], decreasing apoptosis of the infected cells [48] and promoting viral entry into the brain [47]. It has been reported that OPN-N2 may contribute to lymphocyte and neutrophil migration into the brain and decrease type I IFN signaling [12,13,44], whereas OPN-C1 and OPN-C2 decrease apoptosis of the virus-infected leukocytes and other host cells [12]. The blood–brain barrier (BBB), composed of endothelial cell tight junctions and astrocyte extensions, protects the brain against various insults including WNV and other neuroinvasive pathogens. Brain endothelial tight junctions consist of integral membrane proteins (5ccluding, claudins and junctional adhesion molecules) that are involved in intercellular contacts and interactions with cytoplasmic scaffolding proteins, such as zonula occludens (Zo) proteins (Zo-1 and Zo-2), actin cytoskeleton and associated proteins [49,50]. WNV may enter into the brain through multiple pathways, i.e., the direct crossing of the BBB, infection of the endothelial cells of the BBB, infection of olfactory neurons, using the infected leukocytes as transporters and axonal retrograde transport [47,51,52,53,54] (Figure 4). WNV infection induces the expression of many host factors, some of which may directly or indirectly increase the permeability of the BBB, allowing the virus to penetrate to the CNS. MMPs, such as MMP-2 and MMP-9, are found to be potent catalyzers to breach the integrity of the BBB and enhance leukocyte infiltration into the CNS [54,55,56,57]. Brilha et al. have shown that increased MMP-9 activity disrupts the tight junction of the BBB and decreases expression of the tight junction proteins in a coculture model of the BBB [58]. MMP-9 activity is also regulated by an endogenous tissue inhibitor of metalloproteinase-1 (TIMP-1), which preserves the BBB integrity by inhibiting the enzymatic activity of MMP-9 [59]. 

Data from an animal study based on MMP-9 knockout (*Mmp-9^-/-^)* mice showed that MMP-9 mediated WNV entry into the CNS by increasing permeability of the BBB. In *Mmp-9^-/-^* mice, the levels of viral load, inflammatory cytokines and leukocyte numbers were significantly lower than in the wild-type (WT) control group. Moreover, *Mmp-9^-/-^* mice had a higher survival rate following a lethal WNV infection with a tighter permeability of the BBB [54]. OPN has been shown to play a crucial role in MMP-9 activation in melanoma growth and lung metastasis through nuclear factor kappa B inducing kinase or mitogen-activated protein kinase kinase-1-dependent manner [60]. MMP-9 can also increase the biological activities of OPN through PTM. Lindsey et al. showed that OPN could be proteolytically cleaved by MMP-9 in at least 30 sites, and some of the cleaved fragments of OPN could increase cardiac fibroblast migration after post myocardial infarction [61]. Therefore, it is possible that OPN and MMP-9 interaction may facilitate WNV neuroinvasion by disrupting the BBB, which needs further investigation. Previous studies also suggested that OPN is a neuroprotective mediator in traumatic brain injury [62], stroke [63,64] and subarachnoid hemorrhage [65]. Shin et al. showed that in a rat model of experimental autoimmune encephalomyelitis, OPN acted as both a proinflammatory and neuroprotective mediator [66]. The roles of OPN depend on its cellular localization and the stages of the disease course. In the acute stage of the disease, sOPN acts as a chemotactic factor for immune cell migration and is involved in the generation of T helper-1 (Th-1) and Th-17 cells. The iOPN may be involved in the cell survival signaling for recovery in the later stage of disease progression [66]. In addition, other mediators such as MMP-9 may also impact the roles of OPN during the neuroinflammatory stage. 

Researchers from our lab and others have demonstrated that WNV infection increases OPN production in both human and mouse cell culture, blood and brain tissues [47,48]. Importantly, sOPN was also increased in the plasma of the patients during the acute phase of WNV infection and remained at higher levels even after a few years of the recovery, compared to the healthy controls (The Bai Lab unpublished data). To study the possible roles of OPN in the neuroinvasion of WNV, OPN knockout (*Opn^-/-^*) mice were infected with a WT WNV strain (CT2741) via intraperitoneal (i.p.) injection. The *Opn^-/-^* mice displayed relative resistance compared to the WT control (survival rate of 70% vs. 30%), indicating OPN facilitated WNV infection in mice [47]. In one of our earlier studies, we found both human and mouse neutrophils were very susceptible to WNV infection and may serve as carriers for WNV dissemination in the peripheral tissues and the CNS [67]. Followed by WNV infection, *Opn^-/-^* mice had a much tighter permeability of the BBB, less infiltration of WNV-infected neutrophils and lower viral burden in the brain [47]. Moreover, recombinant OPN treatment significantly increased the infiltration of WNV-infected polymorphonuclear neutrophils (PMNs) into the brain, and the mortality rate of *Opn^-/-^* mice following WNV infection [47]. These results suggest that OPN facilitates WNV neuroinvasion by recruiting WNV-infected PMNs into the brain [47]. However, when the viruses were directly injected into the mouse brain bypassing the BBB, *Opn^-/-^* mice produced a slightly higher viral load in the brain on day 6 post infection (p.i.) than the WT controls, suggesting OPN may also have a protective role in the brain against WNV infection [47]. In contrast, another study showed an opposite phenotype of *Opn^-/-^* mice followed by the infection with a mutant strain of WNV (Eg101), which is less neuroinvasive than WT strains due to the lack of an envelope (E) protein glycosylation site [48,68]. After a high dose (10^7^ plaque-forming units) of the Eg101 challenge, *Opn^-/-^* mice presented higher mortality compared to WT mice. In the early time points of the infection, OPN expression was suggested to protect against the viral spread in the CNS by negatively controlling the type I IFN-sensitive, Caspase 1-dependent inflammasome, while promoting an alternative Caspase 8-associated pathway to control the apoptosis of infected cells during WNV Eg101 infection in the CNS [48]. Successful replication of a virus relies on the ability to block or delay apoptosis until sufficient progeny have been produced [69]. OPN has been reported to provide the antiapoptotic signal through binding to α_v_β_3_ [70] and CD44 variants v6 and/or v7 [15,71,72,73]. Thus, it is possible that OPN signaling may help to bypass apoptosis and thereby favor the spread of WNV infection in the CNS. However, this phenotype may also be related to the specific strain of the virus and the overwhelming high dose resulting in the detection of the viruses in the brain as early as day 1 p.i. in both WT and *Opn^-/-^* mice. More studies are needed to dissect the detailed mechanisms of OPN in the pathogenesis of WNV. 

In addition to the acute damages of the CNS, WNV infection has been suggested as a leading factor to cause postinfectious CNS symptoms in patients. The postinfectious CNS symptoms include the proinflammatory state that may contribute to long-term neuroinflammation and autoimmune diseases in WNV survivors [46]. Clinical studies found that some WNV patients developed autoimmune-related diseases, including Myasthenia gravis [74,75,76], Guillain–Barre syndrome (GBS) [77], stiff-person syndrome [78], demyelinating neuropathies [79], brachial plexopathy [80] and poliomyelitis syndrome [79], suggesting that WNV infection may promote or amplify underlying autoimmunity. The postinfectious CNS symptoms are developed due to chronic elevation of various cytokines following recovery from acute illness [46], and OPN is one of these cytokines [81]. OPN has been shown to play critical roles in inducing autoimmune diseases, such as MS, GBS, multiple sclerosis (MS), systemic lupus erythematosus, psoriasis and Sjogren syndrome [82,83,84,85,86]. It could mediate autoimmune diseases through several mechanisms, such as increasing the level of Th-1 and Th-17 cells, dysregulating follicular B cell activation, generating self-reactive autoantibodies, inhibiting apoptosis of autoreactive cells and recruiting leukocytes to the site of inflammation [77]. MG is a long-term neuromuscular disease that leads to varying degrees of skeletal muscle weakness due to autoantibodies that block or destroy nicotinic acetylcholine receptors at the junction between the α-motor neuron and muscle [83,87]. Elevation of OPN during and even after WNV infection may promote the production of autoantibodies that lead to MG. GBS is an immune-mediated demyelinating disease of peripheral nerves, and some patients developed GBS weeks after WNV infection [79]. The concentration of OPN increased in cerebrospinal fluid of GBS patients, which was closely associated with the severity of inflammation of the spinal cord [77]. However, further studies are warranted to determine the roles of OPN in MG and GBS following WNV infection.

## 3. Concluding Remarks

OPN as a functionally diverse molecule has recently received a lot of attention due to its critical roles in tumor progression, inflammation and microbial infections. Despite intensive studies, the detailed roles of OPN are still not well understood. It is partly due to its variety of isoforms, such as OPN-a (sOPN and iOPN), OPN-b, OPN-c and thrombin-cleaved fragments such as OPN-N1 (containing RGD domain), OPN-C1, OPN-N2 (containing SVVYGLR domain) and OPN-C2. The OPN isoforms and fragments may coexist in the same microenvironment and exert different physiological functions, making the investigations of OPN very complicated. In conclusion, OPN plays essential roles in inflammation, wound healing, bone remodeling, tissue debridement, WNV encephalitis and postinfectious CNS symptoms in WNV patients, yet more detailed research is warranted to determine if OPN can be a clinical target for immunotherapeutic interventions.

## Figures and Tables

**Figure 1 vaccines-08-00748-f001:**
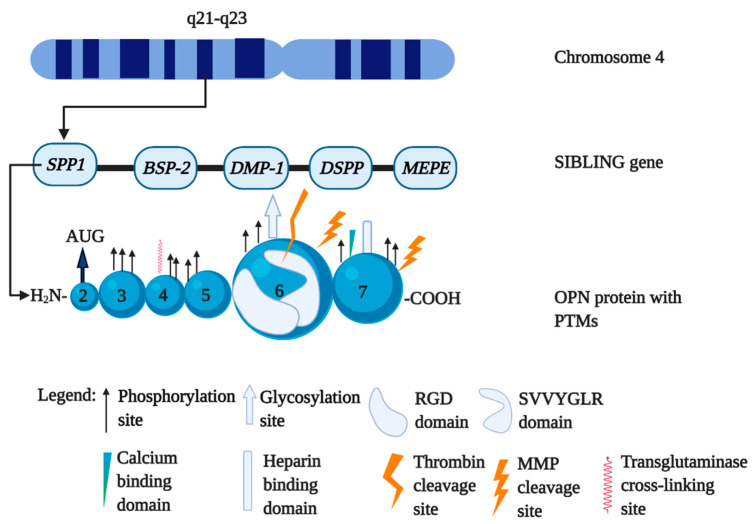
Human *secreted phosphoprotein-1* (*Spp-1*) gene genetic location and the post translational modification (PTM) sites in osteopontin (OPN) protein. OPN gene *Spp-1* is located on human chromosome 4 (4q22.1) at the 5’ end of the cluster of the five genes, collectively called the “SIBLING family”. OPN messenger RNA (mRNA) can be translated into a 314-aa OPN-FL, which is subject to posttranslational modifications (PTMs), such as phosphorylation, O-glycosylation, proteolytic processing, tyrosine sulfation, sialylation, thrombotic cleavage, polymerization and matrix metalloproteinase (MMP) cleavage. The figure was created in Biorender.com.

**Figure 2 vaccines-08-00748-f002:**
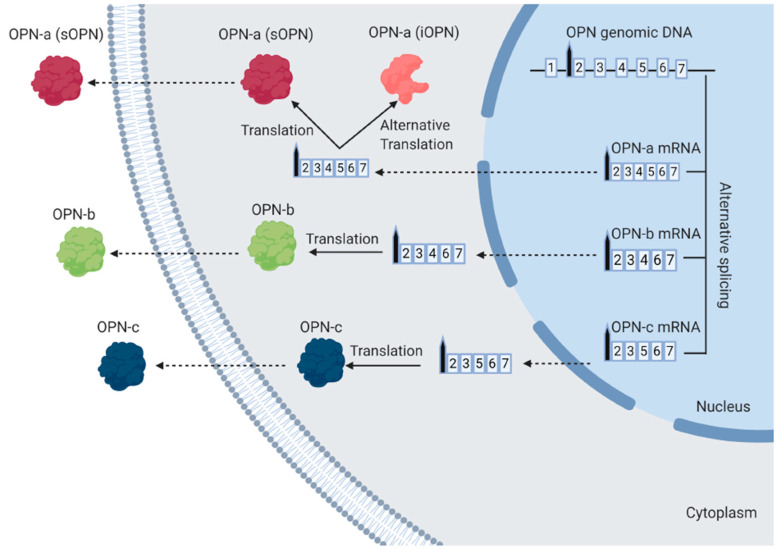
The generation of the different isoforms of OPN. The transcription of OPN genomic DNA produces three different splice variants of OPN mRNA, i.e., OPN-a mRNA, OPN-b mRNA and OPN-c mRNA. OPN-a mRNA contains all of the seven exons, whereas OPN-b mRNA and OPN-c mRNA lack exon-5 and exon-4, respectively. During the translation process in the cytoplasm, OPN-a mRNA produces two distinct isoforms, OPN-a (sOPN) and OPN-a (iOPN) through the AUG or non-AUG translation sites, respectively. The AUG initiates translation to produce OPN-a (sOPN), which is further modified in the endoplasmic reticulum and the Golgi complex and secreted outside the cell. The alternative non-AUG site initiates translation to generate OPN-a (iOPN), which remains inside the cell. OPN-b mRNA and OPN-c mRNA produce OPN-b and OPN-c that are secreted out from the cytoplasm. The figure was created in Biorender.com.

**Figure 3 vaccines-08-00748-f003:**
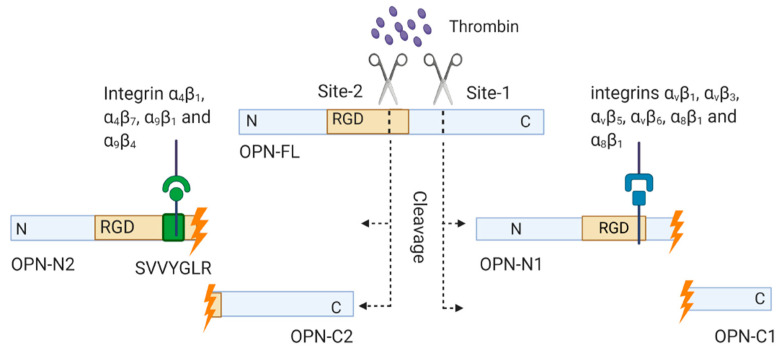
Generation of OPN fragments by thrombin cleavage. OPN can be post translationally modified by thrombin cleavage. Two conserved thrombin cleavage sites have been identified in human OPN. The major site, Site-1 cleavage, produces an N-terminal fragment containing the RGD domain OPN-N1, and a C-terminal fragment OPN-C1. The Site-2 cleavage exposes a cryptic site, the SVVYGLR domain located within the RGD domain. The Site-2 cleavage produces N-terminal fragment, OPN-N2 containing the SVVYGLR domain and C-terminal fragment, OPN-C2. The figure was created in Biorender.com.

**Figure 4 vaccines-08-00748-f004:**
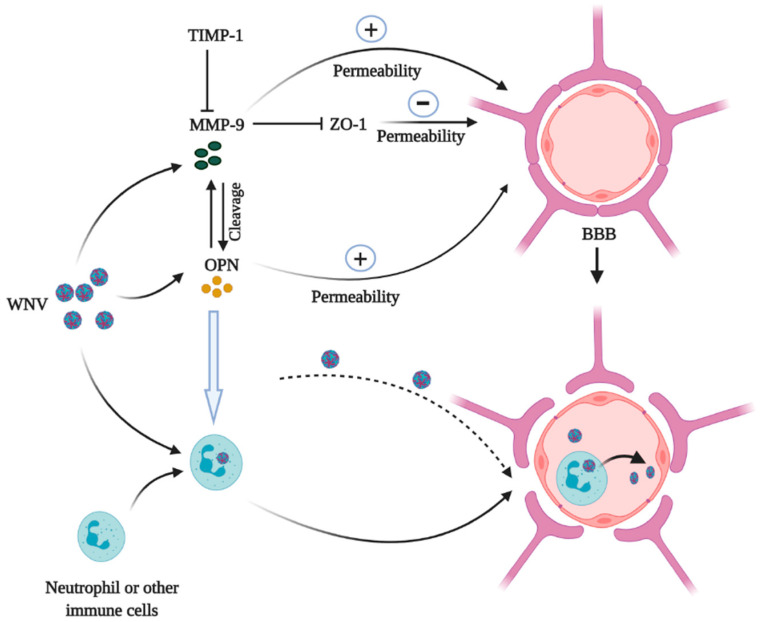
Roles of OPN and MMP-9 in the West Nile virus (WNV) CNS entry. WNV infection induces MMP-9 and OPN expression. MMP-9 stimulates the activity of OPN by PTM cleavage and OPN also stimulates MMP-9 activities. Both OPN and MMP-9 increase the permeability of the BBB during WNV infection. MMP-9 increases the permeability of BBB directly or by inhibiting Zo-1, while TIMP-1 can inhibit MMP-9 activities on the BBB. The increase of the permeability of the BBB facilitates WNV particles to enter into the brain directly crossing the BBB or via neutrophil or other immune cell-mediated “Trojan horse” transportation. Here, OPN increases neutrophil chemotaxis to the site of neuroinflammation. The figure was created in Biorender.com.

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
