# Peer review of "The Roles of Osteopontin in the Pathogenesis of West Nile Encephalitis"

_vaccines, 2020, doi:10.3390/vaccines8040748_

Round 1
Reviewer 1 Report
In this review, the authors discuss the very complicated biology of the multi-functional protein osteopontin and its potential role in the pathogenesis of the Flavivirus West Nile virus. The authors do an excellent job describing all aspects of OPN biology, especially its production in multiple isoforms by a plethora of cell types and their roles in multiple processes, as well as its multiple post-translational modifications. But the major focus of the review is ultimately on the role of OPN in promoting WNV infection in multiple ways, especially its entry into the CNS by crossing the blood brain barrier. This is exemplified primarily by the increased survival (70% to 30%) of OPN knockout mice, likely due in part to their much tighter blood brain barrier permeability, as well as its protection against mechanisms of apoptosis. All of this leads to a potential reduction in WNV-induced post-infection CNS symptoms, including long-term inflammation and autoimmune issues in patients.
This is a well-written and comprehensive review, especially with respect to the aspects of OPN biology. However, with respect to WNV, while it is clear that OPN does play a role in the CNS-related complications of infection by the virus, the mechanisms of this effect are still largely to be determined. Certainly, one can accept that OPN helps WNV transverse the BBB, likely by inducing the expression of several host factors that increase the permeability of the BBB. However, the details of the modification of BBB are still to be elucidated, an issue that the authors make very clear. This is the state of the field and the authors make it clear that there is much to done to clarify exactly how OPN promotes virus infection.
Minor points
Line 241: I believe the authors meant to say facilitated rather than facilities.
Lines 290-2: This is not a complete sentence as written.
Author Response
Response: Thank the reviewer for the suggestion. We have made some modifications on how OPN along with MMP-9 decreases the permeability of BBB and promotes WNV entry into the CNS (Page 7 and Lines 129-146).
Minor points
Line 241: I believe the authors meant to say facilitated rather than facilities.
Response: The error was corrected (Page 8 and Lines 158).
Lines 290-2: This is not a complete sentence as written.
Response: This sentence now is deleted in the revised manuscript.
Reviewer 2 Report
Report:
Farzane Nazneen and Fengwei Bai have submitted a review manuscript entitled “The role of osteopontin in the pathogenesis of West Nile encephalitis”. This is an important topic that needs additional research to determine the role of OPN in regulating blood brain barrier in WNV.
Major comments:
- In general, the introductory part of OPN was about 6 pages where the role of OPN in west Nile virus pathogenesis was only 3 pages. It will be better to make a brief introduction of the OPN as it has been well reviewed in several other publication and discuss more about OPN and WMV interaction.
- Page 5: subsection 1.3.2., the discussion about MMP-9 and OPN is missing.
- In the subsection 1.3.3., the authors are discussing the role of thrombin cleaved OPN. However, I have a hard time to understand the importance of this topic in relation to west Nile virus pathogenesis. The authors need to mention some related hypothesis why this section is important. How WNV impacted on the thrombin cleaved site of the OPN etc.?
- Page 8: OPN has been thought to have an important role in MMP-9 activation. However, the arrow between MMP-9 and OPN shows that MMP-9 is actually activating the OPN contrary to the OPN induces activation of MMP-9. Please clarify.
- Page 9: The authors have shown in their earlier paper and mentioned here that OPN plays a deleterious role in WNV infection by recruiting WNV infected neutrophils to cross BBB. In contrast several studies suggested that OPN has a neuroprotective mechanism and even intracerebral inoculation of OPN improves neurological symptoms, neurological scores, and recovery of BBB function (PMID 25151457, 15678124, 19851092, 18364727). It seems that the function of OPN may be context dependent and the presence of other cytokines and chemokines are equally important to define the function of OPN and its role in regulating BBB during WNV infection. The authors should include the role of other cytokines/chemokines along with the OPN in drawing this conclusion.
- Page 9, lines 277-278: OPN has been shown to induce anti-apoptotic effect by interacting with CD44 and avb3 (PMID: 11342566, 16636021). It will be good to discuss the role of different integrins and CD44 in this context and then explain that OPN has impact on virus infected cell apoptosis.
- Page 10, lines 293-303: WNV infected patients show various neuromuscular diseases including Guillain–Barré syndrome, demyelinating neuropathies, myasthenia gravis, brachial plexopathies, and stiff-person syndrome. The explanation in lines 293-303 seems not relevant to WNV rather a general statement for all other autoimmune disease. The authors need to eliminate that section or cite the relevance to WNV encephalitis.
- The role of TIMP-1, Ang-1, Zo-1 etc. has not been discussed in context to OPN in regulating BBB in west Nile encephalitis. How those proteins will be interacting with OPN needs some schematic presentation.
Minor comment:
- Some of the abbreviations (MMP, RGD) need to be expanded when use for the first time.
Author Response
Major comments:
- In general, the introductory part of OPN was about 6 pages where the role of OPN in west Nile virus pathogenesis was only 3 pages. It will be better to make a brief introduction of the OPN as it has been well reviewed in several other publication and discuss more about OPN and WMV interaction.
Response: We agree. The introductory part of OPN now is condensed. The relevant sections describing WNV pathogenies are modified according to the reviewer’s suggestions.
- Page 5: subsection 1.3.2., the discussion about MMP-9 and OPN is missing.
Response: In the revised manuscript, we reorganized this subsection and moved the necessary discussion on OPN and MMP-9 interaction within the section of WNV pathogenesis. The section is highlighted (Page 7 and Lines 129-146).
- In the subsection 1.3.3., the authors are discussing the role of thrombin cleaved OPN. However, I have a hard time to understand the importance of this topic in relation to west Nile virus pathogenesis. The authors need to mention some related hypothesis why this section is important. How WNV impacted on the thrombin cleaved site of the OPN etc.?
Response: We have updated this section and make it clear in the revised manuscript. The thrombin cleaved OPN fragments can be involved in WNV induced-neuroinflammation. It has been reported that thrombin-cleaved OPN fragment, OPN-N2 might contribute to lymphocyte and neutrophil migration into the brain, decrease type I IFN signaling [1-3], while OPN-C1 and OPN-C2 decrease apoptosis of infected leukocytes and other host cells [1]. Several studies also showed that OPN can provide an anti-apoptotic signal to cells though acting as a ligand for αvβ3 [4] and CD44 variants, v6 and/or v7 [5-9]. The thrombin-cleaved fragments OPN-C1 and OPN-C2 have also been reported to interact with CD44v6 and v7 [1]. The upregulation of CD44v7 protects leucocytes from activation-induced cell death [10, 11] (Page 7 and Lines 127-129).
- Page 8: OPN has been thought to have an important role in MMP-9 activation. However, the arrow between MMP-9 and OPN shows that MMP-9 is actually activating the OPN contrary to the OPN induces activation of MMP-9. Please clarify.
Response: OPN can activate MMP 9 and MMP 9 can also increase the biological activities of OPN. Lindsey et. al. showed that OPN could be proteolytically cleaved by MMP-9 in at least 30 sites and some of the cleaved fragments of OPN can increase the cardiac fibroblast migration after post-myocardial infarction [12] (Page 8 and Lines 154-157). We have revised Figure 4 showing the relationship between OPN and MMP 9.
- Page 9: The authors have shown in their earlier paper and mentioned here that OPN plays a deleterious role in WNV infection by recruiting WNV infected neutrophils to cross BBB. In contrast several studies suggested that OPN has a neuroprotective mechanism and even intracerebral inoculation of OPN improves neurological symptoms, neurological scores, and recovery of BBB function (PMID 25151457, 15678124, 19851092, 18364727). It seems that the function of OPN may be context dependent and the presence of other cytokines and chemokines are equally important to define the function of OPN and its role in regulating BBB during WNV infection. The authors should include the role of other cytokines/chemokines along with the OPN in drawing this conclusion.
Response: Thank the reviewer for raising this important concern. As the reviewer pointed out that OPN has been shown to have a protective role in traumatic brain injury [13], stroke [14, 15], and subarachnoid hemorrhage [16]. In our previous study, we have directly injected WNV into the mouse brain bypassing the BBB and the results showed that Opn-/- mice produced a higher viral load on day 6 post-infection (p.i.) than the wild-type (WT) control in the brain, suggesting OPN may also have a protective role in the brain against WNV infection (reference [17], Figure 2G). This result is consistent with previous reports in this regard. However, when WNV was inoculated via intraperitoneal route, Opn-/- mice had a significantly lower viral burden on day 6 p.i. in the brain compared to the WT mice, indicating the BBB of Opn-/- mice partially blocked the viruses from reaching brain tissues (reference[17], Figure 2G). We now modified the manuscript to discuss these aspects of the role OPN in the pathogenesis of WNV (Page 10 and Lines 184-187).
- Page 9, lines 277-278: OPN has been shown to induce anti-apoptotic effect by interacting with CD44 and avb3 (PMID: 11342566, 16636021). It will be good to discuss the role of different integrins and CD44 in this context and then explain that OPN has impact on virus infected cell apoptosis.
Response: The discussion is included as follows: “Successful replication of a virus relies on the ability to block or delay apoptosis until sufficient progeny have been produced [18]. OPN has been reported to provide the anti-apoptotic signal through binding to αvβ3 [4] and CD44 variants v6 and/or v7 [5-8]. Thus, it is possible that OPN signaling provides protection to bypass apoptosis that may favor the spread of the WNV infection in the CNS” (Page 10 and Lines 195-200).
- Page 10, lines 293-303: WNV infected patients show various neuromuscular diseases including Guillain–Barré syndrome, demyelinating neuropathies, myasthenia gravis, brachial plexopathies, and stiff-person syndrome. The explanation in lines 293-303 seems not relevant to WNV rather a general statement for all other autoimmune disease. The authors need to eliminate that section or cite the relevance to WNV encephalitis.
Response: The section mentioned above is now deleted (Page 11 and Lines 207-226).
- The role of TIMP-1, Ang-1, Zo-1 etc. has not been discussed in context to OPN in regulating BBB in west Nile encephalitis. How those proteins will be interacting with OPN needs some schematic presentation.
Response: We appreciate the reviewer’s suggestion and now the roles of TIMP-1, Zo-1are included in the revised manuscript. The discussion is included as follows: “The blood-brain barrier (BBB) is composed of endothelial cell tight junctions and astrocyte extensions, poses an obstacle for protecting the brain against various insults, including WNV and other neuroinvasive pathogens. Brain endothelial tight junctions consist of integral membrane proteins (occludin, claudins and junctional adhesion molecules) that are involved in intercellular contacts and interactions with cytoplasmic scaffolding proteins, such as zonula occludens (Zo) proteins (Zo-1, Zo-2), actin cytoskeleton; and associated proteins[19, 20]. WNV may enter into the brain through multiple pathways including a direct crossing of the blood-brain barrier (BBB), infection of the endothelial cells of the BBB, infection of olfactory neurons, and uses of infected leukocytes as a carrier or axonal retrograde transport from infected neurons [17, 21-24] (Figure 4). WNV infection induces the expression of many host factors, some of which may directly or indirectly increase the permeability of the BBB allowing the virus to penetrate to the CNS. MMPs, such as MMP-2 and MMP-9, are found to be potent catalyzers to breach the integrity of the BBB and enhance leukocyte infiltration into the CNS [24-27]. Brilha et. al. has shown that increased MMP-9 activity disrupts the tight junction of the BBB and decreases expression of the tight junction proteins in a co-culture model of the BBB [28]. MMP-9 activity is also regulated by an endogenous tissue inhibitor of metalloproteinase-1 (TIMP-1), which preserves the BBB integrity by inhibiting the enzymatic activity of MMP-9 [29] ”. (Page 7 and Lines 129-146). In addition, we also modified Figure 4.
Minor comment:
- Some of the abbreviations (MMP, RGD) need to be expanded when use for the first time.
- Corrections have been made for MMP (Page 5 and Line 80) and RGD (Page 4 and Line 64) accordingly.
References
- Boggio, E., et al., Thrombin Cleavage of Osteopontin Modulates Its Activities in Human Cells In Vitro and Mouse Experimental Autoimmune Encephalomyelitis In Vivo. J Immunol Res, 2016. 2016: p. 9345495.
- Grassinger, J., et al., Thrombin-cleaved osteopontin regulates hemopoietic stem and progenitor cell functions through interactions with alpha9beta1 and alpha4beta1 integrins. Blood, 2009. 114(1): p. 49-59.
- Yokosaki, Y., Matsuura, N., Sasaki, T., Murakami, I., Schneider, H., Higashiyama, S., Saitoh, Y., Yamakido, M., Taooka, Y., Sheppard, D., The integrin α9β1 binds to a novel recognition sequence (SVVYGLR) in the thrombin-cleaved amino-terminal fragment of osteopontin. Journal of Biological Chemistry, 1999. 274(51): p. 36328-36334.
- Song, G., et al., Osteopontin Prevents Curcumin-Induced Apoptosis and Promotes Survival Through Akt Activation via αvβ3 Integrins in Human Gastric Cancer Cells. Experimental Biology and Medicine, 2008. 233(12): p. 1537-1545.
- Iida, T., et al., Is Osteopontin a Friend or Foe of Cell Apoptosis in Inflammatory Gastrointestinal and Liver Diseases? Int J Mol Sci, 2017. 19(1).
- Katagiri, Y.U., Sleeman, J., Fujii, H., Herrlich, P., Hotta, H., Tanaka, K., Chikuma, S., Yagita, H., Okumura, K., Murakami, M., Saiki, I., CD44 variants but not CD44s cooperate with β1-containing integrins to permit cells to bind to osteopontin independently of arginine-glycine-aspartic acid, thereby stimulating cell motility and chemotaxis. Cancer research, 1999. 59(1): p. 219-226.
- Rittling, S.R. and A.F. Chambers, Role of osteopontin in tumour progression. Br J Cancer, 2004. 90(10): p. 1877-81.
- Saleh, S., et al., Osteopontin regulates proliferation, apoptosis, and migration of murine claudin-low mammary tumor cells. BMC Cancer, 2016. 16(1): p. 359.
- Denhardt, D.T., et al., Osteopontin as a means to cope with environmental insults: regulation of inflammation, tissue remodeling, and cell survival. J Clin Invest, 2001. 107(9): p. 1055-61.
- Furger, K.A., et al., The functional and clinical roles of osteopontin in cancer and metastasis. Curr Mol Med, 2001. 1(5): p. 621-32.
- Junaid, A., et al., Osteopontin localizes to the nucleus of 293 cells and associates with polo-like kinase-1. Am J Physiol Cell Physiol, 2007. 292(2): p. C919-26.
- Lindsey, M.L., et al., Osteopontin is proteolytically processed by matrix metalloproteinase 9. Can J Physiol Pharmacol, 2015. 93(10): p. 879-86.
- Chan, J.L., T.M. Reeves, and L.L. Phillips, Osteopontin expression in acute immune response mediates hippocampal synaptogenesis and adaptive outcome following cortical brain injury. Exp Neurol, 2014. 261: p. 757-71.
- Doyle, K.P., et al., Nasal administration of osteopontin peptide mimetics confers neuroprotection in stroke. J Cereb Blood Flow Metab, 2008. 28(6): p. 1235-48.
- Meller, R., et al., Neuroprotection by osteopontin in stroke. J Cereb Blood Flow Metab, 2005. 25(2): p. 217-25.
- Suzuki, H., et al., Protective effects of recombinant osteopontin on early brain injury after subarachnoid hemorrhage in rats. Crit Care Med, 2010. 38(2): p. 612-8.
- Paul, A.M., et al., Osteopontin facilitates West Nile virus neuroinvasion via neutrophil "Trojan horse" transport. Sci Rep, 2017. 7(1): p. 4722.
- Roulston, A., R.C. Marcellus, and P.E. Branton, Viruses and apoptosis. Annu Rev Microbiol, 1999. 53: p. 577-628.
- Luissint, A.C., et al., Tight junctions at the blood brain barrier: physiological architecture and disease-associated dysregulation. Fluids Barriers CNS, 2012. 9(1): p. 23.
- Zhao, Z., et al., Establishment and Dysfunction of the Blood-Brain Barrier. Cell, 2015. 163(5): p. 1064-1078.
- Diamond, M.S. and R.S. Klein, West Nile virus: crossing the blood-brain barrier. Nat Med, 2004. 10(12): p. 1294-5.
- Samuel, M.A. and M.S. Diamond, Pathogenesis of West Nile Virus infection: a balance between virulence, innate and adaptive immunity, and viral evasion. J Virol, 2006. 80(19): p. 9349-60.
- Samuel, M.A., et al., Axonal transport mediates West Nile virus entry into the central nervous system and induces acute flaccid paralysis. Proc Natl Acad Sci U S A, 2007. 104(43): p. 17140-5.
- Wang, P., et al., Matrix metalloproteinase 9 facilitates West Nile virus entry into the brain. J Virol, 2008. 82(18): p. 8978-85.
- Agrawal, S., et al., Dystroglycan is selectively cleaved at the parenchymal basement membrane at sites of leukocyte extravasation in experimental autoimmune encephalomyelitis. J Exp Med, 2006. 203(4): p. 1007-19.
- Esparza, J., et al., MMP-2 null mice exhibit an early onset and severe experimental autoimmune encephalomyelitis due to an increase in MMP-9 expression and activity. Faseb j, 2004. 18(14): p. 1682-91.
- Romanic, A.M. and J.A. Madri, Extracellular matrix-degrading proteinases in the nervous system. Brain Pathol, 1994. 4(2): p. 145-56.
- Brilha, S., et al., Matrix metalloproteinase-9 activity and a downregulated Hedgehog pathway impair blood-brain barrier function in an in vitro model of CNS tuberculosis. Sci Rep, 2017. 7(1): p. 16031.
- Tang, J., et al., TIMP1 preserves the blood–brain barrier through interacting with CD63/integrin β1 complex and regulating downstream FAK/RhoA signaling. Acta Pharmaceutica Sinica B, 2020. 10(6): p. 987-1003.
Round 2
Reviewer 2 Report
The revised paper is much improved. There is no major corrections required.
Author Response
Thank you very much for your very good suggestions that helped us improved the quality of the manuscript.